# Cytosine base editor 4 but not adenine base editor generates off-target mutations in mouse embryos

Hye Kyung Lee [1,4]*, Harold E. Smith[2], Chengyu Liu[3], Michaela Willi [1,4]* & Lothar Hennighausen[1,4]*

Deaminase base editing has emerged as a tool to install or correct point mutations in the genomes of living cells in a wide range of organisms. However, the genome-wide off-target effects introduced by base editors in the mammalian genome have been examined in only one study. Here, we have investigated the fidelity of cytosine base editor 4 (BE4) and adenine base editors (ABE) in mouse embryos using unbiased whole-genome sequencing of a family-based trio cohort. The same sgRNA was used for BE4 and ABE. We demonstrate that BE4-edited mice carry an excess of single-nucleotide variants and deletions compared to ABE-edited mice and controls. Therefore, an optimization of cytosine base editors is required to improve its fidelity. While the remarkable fidelity of ABE has implications for a wide range of applications, the occurrence of rare aberrant C-to-T conversions at specific target sites needs to be addressed.

---

[1] Laboratory of Genetics and Physiology, National Institute of Diabetes and Digestive and Kidney Diseases, US National Institutes of Health, Bethesda, Maryland 20892, USA. [2] Genomics Core, National Institute of Diabetes and Digestive and Kidney Diseases, US National Institutes of Health, Bethesda, Maryland 20892, USA. [3] Transgenic Core, National Heart, Lung, and Blood Institute, US National Institutes of Health, Bethesda, Maryland 20892, USA. [4] These authors contributed equally: Hye Kyung Lee, Michaela Willi, Lothar Hennighausen. *email: hyekyung.lee@nih.gov; science@michaelawilli.com; lotharh@niddk.nih.gov

Deaminase base editing[1,2] directly converts target C·G base pairs to T·A by cytosine base editors (CBE), or target A·T base pairs to G·C by adenine base editors (ABE), without inducing double-stranded DNA breaks[3]. Since the majority of known human pathogenic variants are single-nucleotide alterations[2,4], base editing has been heralded as a high-fidelity tool to correct single-nucleotide polymorphisms (SNPs) associated with many human disorders. While exceptional precision is paramount in a quest to correct somatic and in particular germline mutations, recent studies have revealed that CBEs can induce bystander mutations, including deletions, in mouse zygotes[5] and plants[6]. In contrast, ABE displays a greater fidelity[5,7], even though unexpected C-to-T conversions have been observed with ABE at some target sites[5,8].

Whole-genome sequencing (WGS) of base-edited rice[9] and mouse embryos[10] revealed that BE3, a commonly used CBE, induces a large number of inadvertent base changes throughout the genome, while ABE displays high fidelity. In a separate study, WGS of BE3-edited sheep did not reveal an obvious increase of off-target mutations[11]. Since BE3 can introduce unwanted indels[1,5] and other undesirable base substitutions in addition to C-to-T conversions[5,7,12], the fourth-generation BE4, containing a second uracil glycosylase inhibitor (UGI) domain and optimized linker architectures, appears to have an increased fidelity in vitro[13], in mouse zygotes[5] and in rabbit embryos[14]. Off-target effects for BE4 might be expected based on WGS studies that examined off-target mutations introduced by BE3 and ABE[9,10]. However, as different sgRNAs, editors and computational analytic methods might yield different results[9,10], there is a definitive need for additional studies investigating in vivo genomic effects of BE4 in comparison with ABE at a greater depth. As a note of caution, the original WGS study of CRISPR/Cas9-edited mice suggested the presence of extensive off-target mutations[15], which was, however, likely the result of an imperfect experimental design as pointed out in editorials[16–20]. Further WGS investigations by Iyer and colleagues as well as our group[21,22] using trio studies demonstrated that CRISPR/Cas9 does not introduce an excess of off-target mutations. Therefore, it is scientifically prudent and warranted to examine critical issues of base editing, such as the extent of off-target mutations, with a larger number of mice and under additional conditions. Having several independent studies should provide confidence to those investigators who actively explore therapeutic use in many laboratories and companies. In this study, we have addressed the question of base editing fidelity and conducted unbiased WGS on a total of 44 BE4- and ABE-edited mice, control mice and their wild-type parents, providing more evidence to support previous data and conclusions[9,10].

## Results

**Targeting mouse embryos with BE4 and ABE.** To assess on- and off-target fidelities of the advanced BE4 and ABE in mouse embryos, we conducted a family based trio WGS study (Fig. 1a). Fertilized eggs were injected with BE4 or ABE7.10 mRNA together with a single sgRNA used by both editors, which permitted a direct comparison. Two-cell stage embryos were implanted into surrogate mothers and 13 ABE-edited and nine CBE-edited founder mice were born, together with 13 non-injected controls (Table 1). Tail tissues from 3- to 4-week-old founder mice (Fig. 1b and Supplementary Fig. 1) were screened by Sanger sequencing to identify mutants and targeted deep sequencing was performed to determine haplotypes. ABE introduced A-to-G transitions in the target window and, except one, no bystander or proximal off-target mutations were detected in the 33 edited alleles. In contrast, BE4 induced not only the expected C-to-T transitions but also C-to-G and C-to-A conversions in the target site, frequent proximal off-target mutations, and deletions in four of the nine founders (12 out of the 17 edited alleles) (Fig. 1b and Supplementary Fig. 1). The presence of more than two mutant alleles in some founders is indicative of mosaicism where targeting had also occurred at the two-cell stage, or maybe even later.

**Off-target analysis by WGS.** Unbiased WGS was performed on the 22 edited mice, 13 controls and nine parents at an average depth of 60X (Table 1 and Supplementary Data 1). As non-injected or Cas9-treated embryos don't display significant level of SNVs[10], we used non-injected mice as controls. The WGS data were analyzed using GATK with Joint Genotyping and subsequent filtering to identify single-nucleotide variants (SNVs) and simple indels for each individual mouse (Fig. 2a). Lumpy with SvTyper was used to identify complex and large indels. To explicitly identify de novo mutations located outside the sgRNA, the SNVs and indels present in the parents were subtracted from those identified in the progeny (Supplementary Data 2 and 3). Non-edited control mice had accumulated an average of 132 de novo SNVs (Fig. 2b and Supplementary Data 2). On average 119 de novo SNVs were detected in ABE-edited mice, comparable to that in controls. In contrast, BE4-edited mice carried on average 221 de novo SNVs, a significant increase (Mann–Whitney-U-test: $p = 0.002$), especially C-to-T variants (Fig. 2b, Supplementary Fig. 2 Fig and Supplementary Data 2).

About 2% of off-target SNVs coincided with predicted off-target sites (see M&M for details) suggesting that the majority of mutations were not dependent on the sgRNA and by predicted off-target sites (Supplementary Data 4). The increased off-target editing observed in BE4 but not ABE implies that these mutations were the result of cytosine deaminase AID/APOBEC1 which can induce SNVs in the absence of sgRNAs[1]. C-to-T conversions (plus some C-to-A and C-to-G) are overrepresented in de novo SNVs observed in BE4-edited mice, consistent with enzymatic activity of BE4. Since four out of the nine BE4-edited mice carried additional deletions proximal to the target region, we analyzed globally their indel frequencies compared with controls (Fig. 2c and Supplementary Data 3). The numbers of indels in the BE4- and ABE-edited group showed no differences from the control group (Fig. 2c), also not regarding their characteristics (Supplementary Data 5).

## Discussion

Our results confirm and extend previous work that BE3 but not ABE increases off-target SNVs in mouse embryos[10] and rice[9]. Based on WGS data sets of nine BE4-edited and 13 ABE-edited mice (a total of 50 mutant alleles), we observed a significant mutation rate with the improved BE4, but not ABE, in mouse embryos. While base-edited mouse embryos[10] acquire off-targets independent of sgRNAs, in base-edited rice off-target mutations can coincide with predicted off-target sites[9] suggesting some sgRNA dependence. Here we used the same sgRNA for both BE4 and ABE to eliminate latent sequence-specificity targeting as explanations for off-target mutations in BE4. At this point we cannot assert that off-target effects are due to sgRNA-independence as the TadA enzyme may be much slower to perform its chemistry than APOBEC, resulting in little or no ABE editing at weak Cas9 off-target binding sites. Although different experimental design and analysis methods might lead to different outcomes[9,10], we demonstrated an approximately two-fold increase of de novo off-target mutations in BE4-edited one-cell embryos, which favorably compares to the more than 20-fold increase observed in BE3-edited mouse two-cell embryos[10].

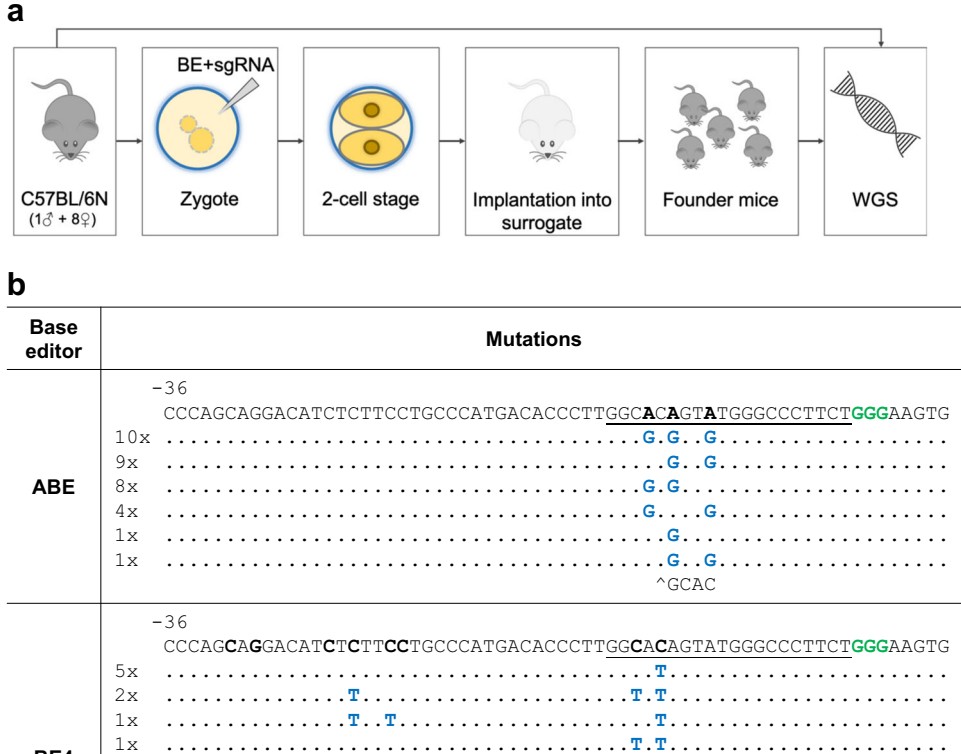

**Fig. 1 Base-editing by BE4 and ABE in mouse embryos. a** Experimental design of the family based trio study. **b** Alignment of sequences from founder mice. 33 mutant alleles edited by ABE, and 17 alleles edited by BE4 were analyzed. The sgRNA sequence is underlined and the PAM sites shown in green. The target nucleotides and edited nucleotide are shown in bold black and blue, respectively. Unintended nucleotide substitutions are shown in red. Deletions are marked with a deletion symbol.

**Table 1 Summary of data obtained from mouse zygotes co-injected with ABE or BE4 mRNA and a sgRNA.**

| No. of Parents | Base editor | No. of injected embryos | No. of embryos implanted | No. of pups born | No. of pups with mutation |
|---|---|---|---|---|---|
| 1 male | Control | – | 68 | 13 | – |
| 8 female | ABE | 196 | 124 | 13 | 13 |
| (~500 pooled zygotes) | BE4 | 170 | 120 | 9 | 9 |

However, given the variation between individual mice, our study and that in rice[9] show a statistically significant difference in off-target SNVs. BE4, an advanced version of BE3, contains an additional UGI (uracil DNA glycosylase inhibitor) and a longer linker, as a means to enhance its specificity[13]. Further studies will be needed to validate the higher genome-wide fidelity of BE4. Notably, BE4 caused inadvertent proximal off-target mutations and deletions in four of the nine founders, which has implications in its use as therapeutic agent. These adverse mutations are likely independent of the sgRNA used as none of the 13 embryos edited with ABE and the same sgRNA displayed proximal off-target mutations and deletions. In addition, although different sgRNAs could influence rates of bystander mutations, previous studies[9,10] show that de novo mutations are induced independent of sgRNAs, and are rather the result of different type of base editors[5]. Since only a few SNVs (<0.01%) coincided with potential off-target sites that had been identified in silico, they are likely the result sgRNA-independent edits. However, it is not clear whether BE3-induced proximal off-target mutations were detected in the two WGS studies[9,10].

In summary, our study emphasizes the high fidelity of ABE as compared to BE4, which also induces increased unwanted base substitutions and deletions in close proximity to the designed target site. Such unwanted mutations are of particular concern when correcting disease-associated SNPs in proteins, as they could adversely alter non-targeted amino acids. Our study also emphasizes the need to monitor off-target mutations in clonal populations, as the analysis of large pools of cells with variable editing, as commonly conducted in in vitro experiments[23–25], results in population averaging. Based on our study and previous experiments[5,9,10], ABE appears to be the current choice for base editing because of its fidelity at target sites and throughout the genome. However, caution about the fidelity of deaminase base editors comes from recent studies that demonstrated extensive off-target RNA editing[26,27] as well as illicit C-to-T conversions introduced by ABE at the target window[5,8].

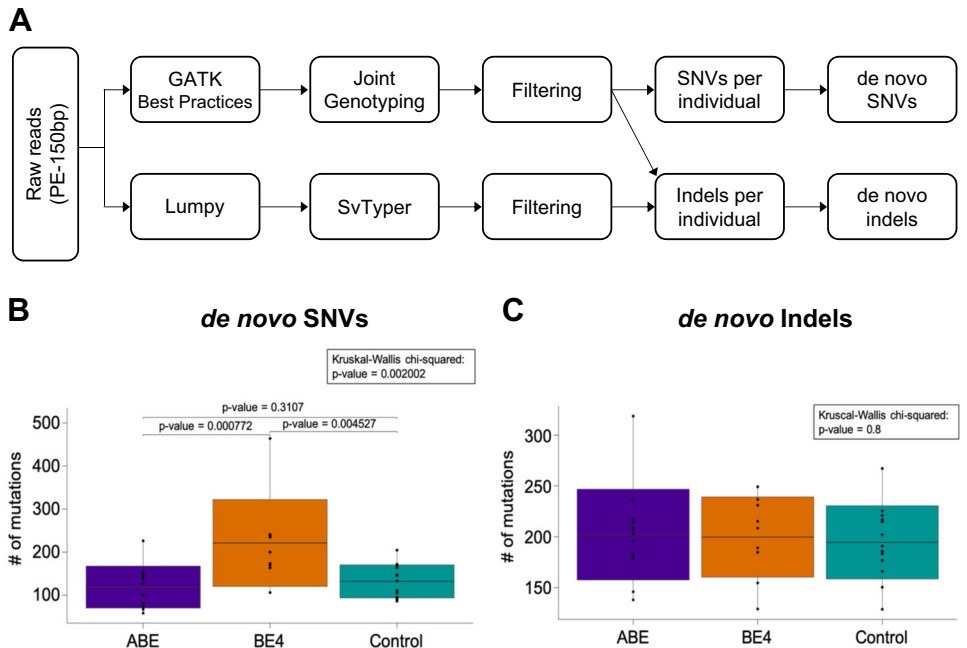

**Fig. 2 De novo mutation frequencies in base-edited and control mice compared to their parents. a** Workflow of whole-genome sequencing analysis for SNVs and indels. **b, c** Numbers of de novo SNVs (**b**) and indels (**c**) identified in mutants (13 ABE-edited mice, $n = 13$; BE4-edited mice, $n = 9$) and controls (non-injected control mice, $n = 13$). Boxplot: center line, mean; box limits, plus and minus standard deviation; whiskers, minimum and maximum. Wilcoxon rank-sum test and Mann–Whitney-U-Test for the pairwise comparison were used to evaluate the statistical significance of differences between groups.

## Methods

**Mice.** All animals were housed and handled according to the guidelines of the Animal Care and Use Committee (ACUC) of the NIH (https://oacu.oir.nih.gov) and all animal experiments were approved by the ACUC of National Institute of Diabetes and Digestive and Kidney Diseases (NIDDK, MD) and performed under the NIDDK animal protocol K089-LGP-17. Base-edited founder mice were generated using C57BL/6N mice (Charles River Laboratories, MD) by the Transgenic Core of the National Heart, Lung, and Blood Institute (NHLBI, MD).

**CRISPR reagents and microinjection of mouse zygotes.** We targeted enhancer 1 within the *Wap* super-enhancer[28] (*Wap* Gene ID: 22373). The Wap-E1 sgRNA (GGCACAGTATGGGCCCTTCT)[28], which contains two cytidines and two adenines near the editing window, was designed and synthesized using ThermoFiser's sgRNA in vitro transcription service. The pCMV-BE4 plasmid (from David Liu's laboratory) and pCMV-ABE7.10 plasmid (Addgene plasmid #102919) were linearized and then their mRNAs were synthesized in vitro using the mMESSAGE mMACHINE T7 kit (ThermoFisher Scientific). Mouse zygotes were produced by in vitro fertilization (IVF) using eggs collected from eight superovulated C57BL/6N female mice and sperm collected from one C57BL/6N male (Charles River Laboratories). The ABE and BE4 mRNAs (50 ng/ul) were separately microinjected with the sgRNA (20 ng/ul) into the cytoplasm of the IVF zygotes. After culturing overnight in M16 medium, those embryos reached 2-cell stage of development were implanted into oviducts of pseudopregnant foster mothers (Swiss Webster, NY). Mice born to the foster mothers were genotyped and subsequently analyzed by WGS.

**Genotyping.** Genomic DNA was isolated from the tip of tails of three to four-week-old founder mice using Wizard Genomic DNA purification Kit (Promega), amplified by PCR, and followed by Sanger sequencing (Quintarabio, CA). Mutations were identified by PCR amplifying a 599 bp fragment encompassing the target sequence, followed by Sanger sequencing. Library preparation and WGS was conducted by the Broad Institute (Cambridge, MA) using Illumina HiSeq X, at a coverage of 60X using 150 bp paired-end reads (Supplementary Data 1).

**PCR Primers.** Wap-S1_F1: GTTGGAACCCATCACAGACAAAGG
   Wap-S1_R1: TGTAGAAACAGAGCAGAGAGGTGG

**GATK analysis.** WGS (60X) was performed on 44 mice, nine parents (one male and eight females), and their progeny, including 22 founder mice carrying base substitutions at target sites induced by ABE or BE4 using one guide RNA and 13 non-injected control ones. The analysis was performed accordingly to the GATK best practices guidelines[29–31] for germline mutations (version 3.8-0). Quality control and alignment was done by BBmap[32] (version 37.36) and BWA MEM[33] (version 0.7.15), respectively, using the reference genome mm10.

For runtime optimization, the aligned BAM files were split up to a chromosome level (for runtime optimization) and reads aligned to different chromosomes were filtered using SAMtools[34] (version 1.5), followed by Picard tools[35] (version 2.9.2) to mark duplicates. The GATK analysis workflow was applied as follows: base recalibration—GATK BaseRecalibrator, AnalyzeCovariates, and PrintReads—using the databases of known polymorphic sites, dbSNP142 and MGPv5 (provided by the high-performance computing team of the NIH (Biowulf)); variant calling—GATK HaplotypeCaller—with the genotyping mode discovery, the ERC parameter for creating gvcf and a minimum phred-scaled confidence threshold of 30. The final step included merging the VCF files of each chromosome (GenomeAnalysisTK, GATK).

**GATK SNV analysis.** Joint genotyping was applied on all 44 samples together and hard filters were applied: QD < 2.0 || FS > 60.0 || MQ < 40.0 || MQRankSum < −12.5 || ReadPosRankSum < −8.0 || SOR > 3. The resulting SNVs were additionally filtered by removing those overlapping with repetitive elements[36] (UCSC's masked repeats plus simple repeats; http://hgdownload.soe.ucsc.edu/goldenPath/mm10/database) and black regions (ENCODE[37]; http://mitra.stanford.edu/kundaje/akundaje/release/blacklists/mm10-mouse/). On an individual level, only SNVs with a genotype of 0/1 or 1/1 were kept. Further filtering steps comprised the removal of SNVs with a read depth smaller than 10, an excessive read depth[38] ($d + 3\sqrt{d}$, where $d$ is the average read depth), an allele frequency less than 10% using a variety of tools (BEDtools, version 2.26.0; BEDOPS, version 2.4.3; VCFtools, version 0.1.17)[39–41]. All SNVs within ±5 bp of an indel border were removed as likely false-positives.

**Simple GATK indel analysis.** Indels identified by GATK where extracted after joint genotyping and subsequently hard filters were applied according to the GATK recommendations: QD < 2.0 || FS > 200.0 || ReadPosRankSum < −20.0 || SOR > 10.0. Indels overlapping with repetitive elements[36] (UCSC's masked repeats plus simple repeats; http://hgdownload.soe.ucsc.edu/goldenPath/mm10/database) or black regions (ENCODE[37]; http://mitra.stanford.edu/kundaje/akundaje/release/blacklists/mm10-mouse/) were removed. The individual samples were filtered keeping only indels with the genotypes of 0/1 and 1/1, removing those with a read depth smaller than 10 as well as sites with an excessive number of reads[38] ($d + 3\sqrt{d}$, $d$ = average read depth). Last, all simple indels that overlap with complex indels identified using LUMPY (version 0.2.13) were excluded. For all those steps a variety of tools[39–41] was applied.

**Complex indel analysis using Lumpy.** The analysis of complex indels was done on the same samples using Lumpy[42] according to the guidelines. Mapping was done using BWA MEM[33], with the parameters –excludeDups –addMateTags –maxSplitCount 2 –minNonOverlap 20 (reference genome mm10), followed by Lumpy[42] using the discordant and split reads as input and genotypes were

identified using SVTyper[43]. The subsequent filtering steps comprised the selection of indels with a genotype of 0/1 and 1/1 and the removal of indels with a quality smaller than 100 and an excessive read coverage ($d + 3\sqrt{d}$[38], where $d$ is the average read depth) or a SU value (Number of pieces of evidence supporting the variant across all samples) smaller than 5. Indels overlapping with repetitive elements[36] or black regions[37] were excluded.

**Statistics and reproducibility**. All statistical analyses for 13 non-injected control, 13 ABE-edited and 9 BE4-edited mice were performed with R package 3.3.3 (http://www.R-project.org/). Kruskal-Wallis test was applied using kruskal.test and pairwise comparison was done with a Wilcoxon Rank Sum wilcox.test in R. All values represent means ± S.D.

**Targeted deep sequencing**. Target sites were amplified from mouse genomic DNA using Phusion polymerase (Thermo Fisher Scientific) and PCR products were prepared as libraries for next-generation sequencing. Pooled PCR amplicons were conducted paired-end sequencing using an Illumina MiSeq (Illumina).

**PCR primers**. Wap-S1_F2: AAGACAGGAGGTTTTGAGCAAGGC
Wap-S1_R2: CACCAGTGAAGACAAAGGAGTATGG

**Off-target analysis**. Off-target sites were predicted using CRISPOR http://crispor.tefor.net/ [44]. The resulting off-target sites were filtered using the same criteria as for SNVs and indels, to consider only those areas of the genome which do not coincide with black regions[37] (ENCODE[37]; http://mitra.stanford.edu/kundaje/akundaje/release/blacklists/mm10-mouse) or repetitive elements[36] (UCSC's masked repeats plus simple repeats; http://hgdownload.soe.ucsc.edu/goldenPath/mm10/database). Mutations, which were present in the population and not only in base-edited mice, but identified at predicted off-target sites, were not considered as a consequence of base editing.

**Reporting summary**. Further information on research design is available in the Nature Research Reporting Summary linked to this article.

## Data availability
The data are available at SRA under project number PRJNA555149.

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

## Acknowledgements
This work utilized the computational resources of the NIH HPC Biowulf cluster (http://hpc.nih.gov). This work was supported by the Intramural Research Programs (IRPs) of National Institute of Diabetes and Digestive and Kidney Diseases (NIDDK) and National Heart, Lung, and Blood Institute (NHLBI).

## Author contributions
All authors designed the study. C.L. generated mutant mice. H.K.L. identified and characterized mutant mice and performed experiments and data analysis. H.E.S. and

M.W. performed computational analysis. H.K.L., M.W., and L.H. supervised the study and wrote the manuscript and all authors approved the final version.

## Competing interests

The authors declare no competing interests.
