## [Peer Review File · Communications Biology]

Editorial Note: *This manuscript has been previously reviewed at another Nature Research journal. This document only contains reviewer comments and rebuttal letters for versions considered at Communications Biology.*

REVIEWERS' COMMENTS:

Reviewer #1 (Remarks to the Author):

The manuscript reports on the use of whole genome sequencing to compare the fidelity of the cytosine base editor 4 (BE4) and adenine base editor (ABE7.10) in mouse embryos. Understanding the precision of these tools is important because of their widespread use in research and their therapeutic potential.

My overall impression from having seen the original and revised submissions is that the work is convincing and the genome-editing field will benefit from their findings. While reproducibility with bioinformatics-based projects remains a challenge, the sequences are available and there seems to be enough detail on the analysis tools.

Minor comment:

Given that the experiment of performing a no-sgRNA control has come up repeatedly, it would be useful if the authors acknowledged this specifically as a caveat in their discussion section (while citing other studies as providing this data already). The readers will ask as well.

Reviewer #2 (Remarks to the Author):

The manuscript I believe is appropriate for publication in Communications Biology. I have a few minor comments:

Line 92 has a typo: "To explicitly identify de novo mutations that are excluded ones within sgRNA"

Line 100: "About 2% of off-target SNVs coincided with the predicted off-target site, CRISPR, suggesting that the majority of mutations were not dependent of the sgRNA and predictable by predicted off-target sites" The authors should cite their off-target predictor tool. Is the name of the tool (CRISPR) a typo? I have not heard of this one, do they mean CRISPOR?

Line 120: " Here we used the same sgRNA for both BE4 and ABE and therefore eliminate latent promiscuous guide targeting as the explanation for off-target mutations in BE4." The authors cannot make this statement without conducting the proper experiment (BE4 and ABE without gRNA), which they state in their rebuttal would take significant time and resources to complete. If this is so, then they need to modify this sentence and any others where they assert that their off-target effects are due to gRNA-independence. The TadA enzyme may be much slower to perform its chemistry than APOBEC, resulting in no ABE editing at weak Cas9 off-target binding sites. The off-target prediction algorithms for Cas9 are known to be fallible, so just because certain mutations do not occur at predicted off-target sites given the gRNA sequence.

The authors also could analyze their data and compare where the off-target mutations are occurring among the different BE4-treated mice. If the off-target edits are occurring at different sites in different

mice this would indicate gRNA-independent activity. However, if they occur at the same sites it could be due to sequence-specificity of the deaminase, DNA accessibility reasons, or gRNA-dependence.